# Effects of Mental Fatigue on Reaction Time in Sportsmen

**DOI:** 10.3390/ijerph192114360

**Published:** 2022-11-02

**Authors:** Gian Mario Migliaccio, Gloria Di Filippo, Luca Russo, Tania Orgiana, Luca Paolo Ardigò, Marcela Zimmermann Casal, Leonardo Alexandre Peyré-Tartaruga, Johnny Padulo

**Affiliations:** 1Sport Science Lab, 09131 Cagliari, Italy; 2Department of Psychology, Niccolò Cusano University, 00166 Rome, Italy; 3Department of Human Sciences, Università Telematica degli Studi IUL, 50122 Florence, Italy; 4Department of Teacher Education, NLA University College, Linstows Gate 3, 0166 Oslo, Norway; 5Department of Neurosciences, Biomedicine and Movement Sciences, School of Exercise and Sport Science, University of Verona, via Felice Casorati 43, 37131 Verona, Italy; 6LaBiodin Biodynamics Laboratory, Universidade Federal do Rio Grande do Sul, Porto Alegre 90690-200, Brazil; 7Department of Biomedical Sciences for Health, Università degli Studi di Milano, 20134 Milan, Italy

**Keywords:** heart rate, mental fatigue, response time, Stroop test

## Abstract

Aim: Mental fatigue (MF) has been defined as a psychobiological state commonly caused by prolonged periods of demanding cognitive activity. However, the differences between women and men in their reaction times (RTs) to visual stimuli due to mental fatigue remain largely unknown. We compare the differences in RT and heart rate after an acute intervention of mental fatigue between male and female athletes. Materials and methods: For this aim, 64 participants (age 31.7 ± 6.2 y) performed a routine of 15 min of the Stroop test (PsyTool), with 600 tasks and five different colors. Their heart rate (HR) was registered before, during, and one, three, and five minutes after the Stroop test. Meanwhile, the RT was evaluated before and after the Stroop test. A general linear mixed model (GLMM) and a Bonferroni post hoc test were used to compare the HR between the conditions and an ANOVA two-way analysis was used to compare the values pre-/post-Stroop test. (α = 0.05). Results: The GLMM for HR showed an effect on the time (*p* < 0.001) and the time × group interaction (*p* = 0.004). The RT was significantly increased pre- to post-Stroop test (*p* < 0.05); however, there was no difference between the pre- and post-HR measurements (*p* = 1.000) and the measurements one (*p* = 0.559), three (*p* = 1.000) and five (*p* = 1.000) min after the Stroop test. Conclusion: The present findings suggest that the parasympathetic branch of the autonomous nervous system which functions as a relaxation system tends to be activated under increasing mental fatigue, with a decreased performance (RT) similarly in men and women. Therefore, athletes could use MF induced during training to improve the time delay related to motor tasks.

## 1. Introduction

Mental fatigue (MF) is defined as a decline in one’s psychobiological state caused by demanding and prolonged periods of cognitive activity [1,2,3]. MF is a common occurrence that exists in daily tasks but has garnered attention from the scientific community regarding its effects on cognitive performance and attention [2]. More recently, new evidence suggests that MF extends beyond its impact on cognitive functioning and impairs subsequent performance during prolonged and submaximal exercise [3]. MF can affect human performance and is considered critical in the design and evaluation of complex human–machine systems; therefore, it can contribute to the alteration of some elements of motor response to visual stimuli (eye–hand) with significant effects on reaction time (RT). RT provides an indirect index of the central nervous system’s processing capacity and a simple means of determining sensorimotor performance [4]. In the world of work, higher RT times have been observed in shift workers; rotating night shift workers have longer visual reaction times than day workers [5]. In elite sports, various examples are applicable: the blocks in athletics or swimming, the jab in fencing and the shotgun in skeet shooting. Similar situations can be found in all sports in which you must adapt quickly to changing situations [4].

The decreases in performance attributed to mental fatigue were particularly pronounced in tasks involving voluntary attention control. Maintaining high levels of performance for prolonged periods of time involves high cognitive effort characterized by high levels of alertness, selective attention, decision making, and automated control mechanisms (such as hand–eye) and can contribute to a high level of fatigue [6]. Furthermore, performing repeated tasks can contribute to the increase in muscle fatigue with an associated induction of mental fatigue [7]. Among these, some tasks that require high attentional vigilance but low neuromuscular work with limited muscle contractions can induce a sense of effort and fatigue; subsequently, cognitive factors and mental stress can cause muscle fatigue [8]. Although mental fatigue is a common and daily occurrence, the literature on mental fatigue is almost exclusively limited to its effects on cognitive performance and attention. More recently, evidence suggests that the effect of mental fatigue extends beyond its impact on cognitive functioning and impairs subsequent exercise performance [3].

The purpose of this investigation was therefore to expand this research and investigate the effect of mental fatigue on physiological, psychological and performance variables during a hand–eye reaction time task as well as any mechanisms underlying the effect. The reference horizon is the effect of MF on performance in sports that include a particularly short response time to visual stimuli in their performance models. Similarly in a healthy population and in a population of elite athletes, reaction times are necessary for motor responses following a visual stimulus. Both populations are subject to mental fatigue, and both must respond to the increasingly frequent stresses of their surrounding environments and to the strong pressures associated with competitive events. Mental fatigue has become a widespread sub-health condition and severely impacts the brain’s cognitive function. Data suggest that brain activity in a mentally fatigued state is largely different from resting and task states [9]. MF has been defined as a psychobiological state caused by prolonged periods of demanding cognitive activity and is characterized by subjective sensations that are perceived as “fatigue” or a “lack of energy” [2]. The effects of mental fatigue have been extensively studied in various aspects of the cognitive performance of pilots and healthcare workers [10]. However, the impact of mental fatigue on the subsequent execution of motor reaction times to visual stimuli remains largely unknown. The main aim of the present study was to experimentally observe the hypothesis that mental fatigue impairs reaction times in healthy and physically active people and that interference could also be detected in additional physiological markers. To test this hypothesis, we measured the reaction times before and after 15 min of mental fatigue induced by the Stroop test. The Stroop test is a cognitive task that requires a high degree of sustained attention, response inhibition and error monitoring, and has been used several times before to induce a state of mental fatigue [11,12,13,14]. However, MF can independently influence the control of the autonomic nervous system, resulting in an increased cardiovascular effort without a simultaneous physical task. This aspect runs counter to the idea that the influence of mental fatigue can be confined to the realms of relevant cognitive performance; it affects the perception of fatigue on effort and associated cardiovascular responses [15]. The autonomic adaptations to exercise are in fact mediated by central signals coming from the upper brain (central command) and by a peripheral reflex deriving from the work of the skeletal muscles (exercise pressure reflex), with further modulation provided by the arterial baroreflex [16]. In the specific case of sports performance, the goal of victory and defeat is closely linked to cardiovascular responses. It has been seen that in the presence of mental fatigue, cardiovascular responses increase in the presence of conditions that present a greater probability of victory. Conversely, responses tend to weakly decrease when the odds of winning remain low [17]. Therefore, we asked the participants to pay close attention to the task, its accuracy, and its execution time.

## 2. Materials and Methods

### 2.1. Participants

Sixty-four participants of both sexes (34 females: age 34.43 ± 7.41 years, body mass 56.64 ± 8.42 kg, body height 162 ± 8.22 cm; and 30 males: age 29.38 ± 3.28 years, body mass 78.71 ± 6.97 kg, body height 175 ± 7.84 cm) recruited from a national sports center took part in the present study. All subjects were right-handed except two participants (one male and 1 female, who were left-handed). The inclusion criteria for participation in this study were: level of physical activity higher than 300 min per week for a continuous period of three years according to the international guidelines [18]; in good health without muscle, nerve or tendon injuries; free from the use or influence of drugs and in the absence of drugs except contraceptives for the female population; and regular sleep of no less than 7 h. All subjects gave their written informed consent to participate in the study after receiving a thorough explanation of the study’s protocol. The protocol conformed to internationally accepted policy statements regarding the use of human participants in accordance with the Declaration of Helsinki and was approved by the Departmental Research Authorization Committee of Niccolò Cusano University.

### 2.2. Experimental Setting

Subjects abstained from drinking alcohol or caffeinated beverages for 24 h prior to the test and fasted for at least 2 h prior to the experimental procedure to reduce any nutritional interference with this investigation. Training status was checked and none of the participants underwent endurance and/or resistance activities outside their normal training protocol for this investigation. The data were acquired between October and November 2021 in the same time slot from 4 to 7 p.m. in a sports science laboratory with a temperature and relative humidity for each session ranging from 22 to 24 °C and 4 to 15%, respectively. There were two visits with a 48-h interval for procedures (Figure 1). In the first visit, all participants were familiarized with both the reaction time and the Stroop test; while in second session, the experimental protocol was proposed to all participants. 

Reaction time (RT) test was performed by each participant with three trials separated by an interval of 30 s. The participants were instructed to keep their index finger of their prevailing hand (right or left for left-handed participants) in contact with the spacebar. Upon seeing the visual stimulus on a digital screen, the participant pressed the spacebar as quickly as possible. Cognitive Fun test (http://cognitivefun.net/test/1, accessed on 15 March 2021) with a sampling rate of 500 Hz was used for RT test. The distance from the screen to the eyes of the subjects was set at about 55 cm. The software displayed the time in milliseconds on the screen, which was immediately reported on an Excel data sheet. Reaction time is an important indicator of attention and consists of the time taken to respond to an external stimulus. In this case, the participants were instructed with the index finger of their dominant hand to press the spacebar of a keyboard as soon as a green spherical shape (10 cm in circumference) appeared on the notebook screen.

Stroop test provided by PsyToolkit, an open-source software for psychological testing, was carried out, with several successive tests of 200 questions out of a total of 600 in fifteen minutes. The participants were instructed to keep their index finger of their prevailing hand (right or left for left-handed participants) in contact with the spacebar and to move their index finger following the scheme showing the word/color combination for the 5 foreseen combinations (Figure 2):C key combined with yellow colorV key combined with red colorButton B combined with green colorN button combined with blue colorM button combined with white color

On the second trial, each participant took part in the test without interruptions, as follows: 

Reaction Test PRE → Stroop Test → Reaction Test Post → Rest 5’

The participant was instructed to keep a heart rate monitor belt for the entire duration of the test to allow reading of the data in several moments of the protocol:T0—At the initial trialT1—At the end of the Reaction test (PRE)T2—After 200 questions of the Stroop testT3—After 400 questions of the Stroop testT4—At the end of the Stroop test, or 600 questionsT5—At the end of the Reaction test (POST)T6—At rest, after 1’T7—At rest, after 3’T8—At rest, after 5’

The heart rate was monitored during all test conditions using a Polar H7 belt-type heart rate monitor (Polar Electro Oy, Kempele, Finland). Resting heart rate tests were performed after the participant was lying on their back with arms at their sides and with controlled breathing on a horizontal bench in a room with constant temperature, no wind and limited sound. At the end of the five minutes, the heart rate was checked by reporting the lowest point in the next 30 s. The heart rate reading was performed with the athletes in a sitting position, with their hands on the keyboard. All HR data were then normalized for each condition as a percentage of HR_MAX_. FC_MAX_ was obtained from the formula 208 − 0.7 × age [19].

All variables measured were as follows:Heart rate (HR)Resting heart rate (HR_R_)Maximum heart rate (HR_MAX_)Reaction time (RT)Heart Rate after Reaction Time before Stroop test (HR_PRTR_)Heart Rate after Reaction Time after Stroop test (HR_POTR_)Reaction time before the Stroop test (RT_PR_)Reaction time after Stroop test (RT_PO_)

### 2.3. Statistical Analysis

All descriptive measures are mean values ± SD. Shapiro–Wilk test was applied to check the normality of the data. The HR was analyzed using the generalized linear mixed model (GLMM), with the comparison between the groups (male × female) and moments of data collection (T0 × T1 × T2 × T3 × T4 × T5 × T6 × T7 × T8). Firstly, intra-subject variability was tested to define candidate random variables due to the hierarchical nature in GLMM [20]. The condition was not found to be a variable according to the test of compliance with the intraclass correlation coefficient (ICC-pre) of the analysis of variance components by the maximum restricted likelihood approach. The pre-ICC was not higher than 5%, and none had a random effect. The TR was analyzed using a two-way ANOVA, with the comparison between the groups (male × female) and moments (pre × post). The Bonferroni post hoc test was used to identify the differences between effects and interactions. A significance level of ɑ = 0.05 was adopted and all data were analyzed in the Statistical Package for the Social Science (SPSS), version 27.0 (SPSS Inc., Chicago, IL, USA).

## 3. Results

All participants (*n* = 64) completed each trial session regularly. The Shapiro–Wilk test showed that the HR and RT data showed a normal distribution (*p* = 0.975). A summary of the data can be seen in Table 1.

The descriptive statistics (Figure 3) show that the estimated maximum heart rate (HR_MAX_) was found to be 185.8 ± 4.3 bpm (*p* = 0.029 between the two groups, 187.6 ± 2.2 and 184.3 ± 5.1 bpm for males and females, respectively), while the HR_R_ was found to be 35.1 ± 5.4 %HR_MAX_ (*p* = 0.790 between the two groups, 34.9 ± 6.0 and 35.2 ± 4.9%HR_MAX_ for males and females, respectively). HR_PRTR_ was found to be 38.6 ± 7.6 %HR_MAX_ (*p* = 0.408 between the two groups, 37.8 ± 8.1 and 39.4 ± 7.2%HR_MAX_ for males and females, respectively), while HR_POTR_ was found to be 38.5 ± 7.1%HR_MAX_ (*p* = 0.615 between the two groups, 39.0 ± 7.8 and 38.1 ± 6.5%HR_MAX_ for males and females, respectively).

The generalized linear mixed effects model (GLMM) for HR showed an effect of time (Fischer value of 33.012 and a η^2^ 0.347 with *p* = 0.001) and the time * group interaction (Fischer value of 1.101 and a η^2^ 0.017 with *p* < 0.361). The Bonferroni post hoc test showed a significant difference between the measurements at rest (T0) and all other measurements (*p* < 0.05). A difference was observed during the Stroop test in relation to the other conditions (*p* < 0.05); however, there was no difference between the HR_PRTR_ and HR_POTR_ measurements (*p* = 0.905) and the measurements after one (*p* = 0.559), three (*p* = 0.998) and five (*p* = 0.905) minutes. 

Furthermore, the descriptive statistics relating to the RT (Figure 4) show that the RT_PR_ (pre-Stroop test) was found to be 308.3 ± 40.1 ms (*p* = 0.128 between the two groups, 316.5 ± 39.9 ms and 301.1 ± 39.4 ms for males and females, respectively) while the RT_PO_ (post- Stroop test) was found to be 334.1 ± 40.7 ms (*p* = 0.60 between the two groups, 344.3 ± 39.9 and 325.1 ± 39.9 ms for males and women, respectively). The analysis of variance (ANOVA) for RT showed a Fischer value of 64.303 and a η^2^ 0.503 with *p*< 0.001, while the post hoc test showed a significant difference between RT_PR_ and RT_PO_ (*p* = 0.000), both for males and females, but with no significant difference between the groups (*p* > 0.05).

## 4. Discussion

The aim of this study was to investigate the effect of mental fatigue on reaction times in subjects with a high level of fitness. It was hypothesized that mental fatigue would most affect performances in sports activities that require a rapid motor response to a visual stimulus in hand–eye tasks. The literature has shown that some sports athletes have a better ability to use visual signals of their opponent’s movements than less qualified athletes. In table tennis, for example, quick processing allows them to be more successful in predicting the future trajectory of the ball.

In addition, faster reactions offer additional time to perceive one’s environment and plan proper movements. It is noted how athletes, in achieving their best performance, must use their perceptive, cognitive and motor resources to the maximum to produce movements as quickly as possible in response to a visual stimulus. What is certainly a key factor in reducing motor response times is the absence of mental fatigue, which is responsible for a worsening of performance in terms of RT.

RT can be considered a decisive factor for victory during competitions in numerous Olympic sports. In table tennis in particular, the incredible speed of the ball and the short distance it travels between the opponents requires a minimal reaction time to execute shots. Thus, tennis players barely have time to react to the visual stimulus of the ball before the ball arrives to be hit. In this sport, athletes must accurately anticipate their opponent and have constant vigilance; they must promptly react to the sound of the ball as well as to the precise biomechanics of the shot, which allows them to select and execute the motor scheme with the best possibility to win the match. 

Traditionally, the measure of RT has been used to indicate athletes’ ability to elaborate in terms of decision-making. In this study, the descriptive data of simple RTs demonstrated that subjects with induced mental fatigue have a significant increase in response times. In addition, a simultaneous increase in heart rate with progressive kinetics was observed during the fifteen minutes of the Stroop, and then it returned to baseline levels at the fifth minute of rest. These results are interpreted as proof that simple mental fatigue induced for fifteen minutes can alter the athlete’s psychological and physiological state. However, the test of reaction times carried out pre- and post-fatigue was only able to differentiate the measure of the hand–eye response times but was not sufficiently calibrated to the specific effects of intensive training on athletes or on specific sports’ skills. It is interesting to note that when the athlete undergoes mental fatigue in a competition context, the RT values are significantly higher than the baseline level. As demonstrated in endurance sports [3], it is of interest to suggest a further effect on sports that require the rapid processing of complex stimuli in the shortest possible time. This confirms the reasoning that maintaining athletes’ mental fatigue is highly correlated with complex and specific tasks. Furthermore, the variation in heart rate that occurs in the total absence of movement is explained by an activity of the sympathetic nervous system, a further limiting factor in the performance of sports requiring a low RT. The response time test could further distinguish the performance of athletes and non-athletes, as well as amateur and advanced athletes, which was not detectable by this study. What was particularly interesting in this experiment was the fact that mental fatigue appears to play a role during rapid and complex motor response phases, such as the conditions that athletes undergo in natural competitions. The test chosen for this experiment, the 15 min Stroop task with 600 questions, allowed us to first report a significant difference in RT. However, it is not unusual to find studies that report different types of tests and overall durations that allow researchers to obtain significant alterations in RTs. In short, the present study used the Stroop test as a suitable tool to create induced mental fatigue that allowed us to recreate the mental state of athletes engaged in competition actions. It is worth noting that in sports characterized by a low RT, such as fencing [21], table tennis [22] and shooting [23,24,25], players are required to have temporal and spatial precision to be able to hit the primary target: the opponent, ball or an object in flight. The choice to evaluate the effect of mental fatigue allowed us to think about the trade-off that occurs in natural environments in which speed and spatial precision are correlated with the level of skill and RT times are decisive for achieving the optimal result.

## 5. Conclusions

In summary, the present study used induced mental fatigue tasks to investigate the effects on motor reaction times, especially in hand–eye tasks. The results of this study provide experimental evidence that mental fatigue limits performance similarly between women and men by increasing motor response times to a visual stimulus in the presence of an increased cardiovascular load. These data have various implications.

First, it suggests that MF has a direct relationship to RT and that its alteration can become a determining factor in sports performances. Secondly, it suggests that the detection of HR can be used as a verification marker of mental fatigue, in the absence of physical fatigue. Furthermore, the test employed can be used as a tool to improve RT for mental fatigue conditions. Therefore, our results can be used for the training of athletes to recreate the expected competition conditions in natural environments. Therefore, we can conclude that: (1) the RT of athletes subjected to mental fatigue is higher than that of those in resting conditions; and (2) the HR of athletes subjected to mental fatigue is higher than in that of those in resting conditions. Further research should investigate whether the cardiovascular response in the presence of mental fatigue is subject to states of performance anxiety, which is particularly frequent in athletes.

## Figures and Tables

**Figure 1 ijerph-19-14360-f001:**
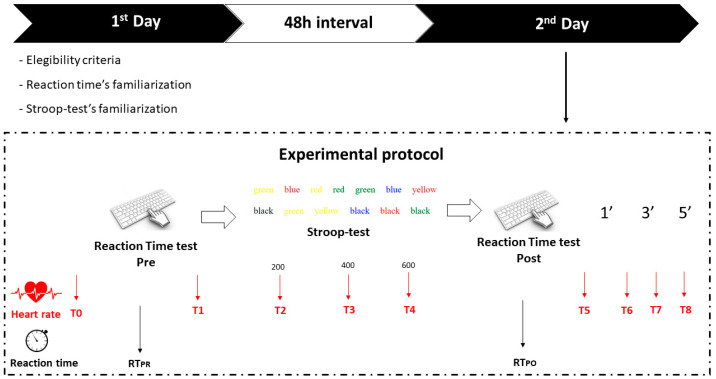
Study design.

**Figure 2 ijerph-19-14360-f002:**
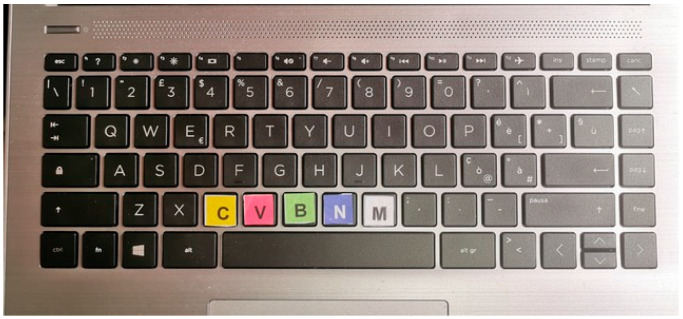
Color modified keyboard.

**Figure 3 ijerph-19-14360-f003:**
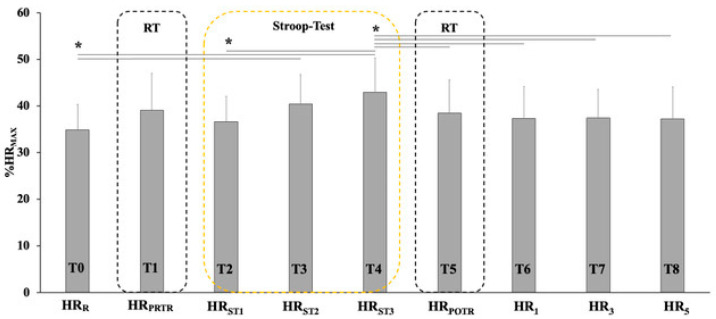
Heart rate in each condition of the experimental test. HR_R_: Frequency at rest; HR_PRRT_: Pre-Stroop test heart rate; HR_POTR_: Post-Stroop test heart rate; HR_ST1_: Heart rate at 1/3 of the Stroop test; HR_ST2_: Heart rate at 2/3 of the Stroop test; HR_ST3_: Heart rate at the completion of the Stroop test; HR_1_: Heart rate 1 ‘from the end; HR_3_: Heart rate 3 ‘from the end; HR_5_: Heart rate 5 ‘from the end. *—*p* < 0.05.

**Figure 4 ijerph-19-14360-f004:**
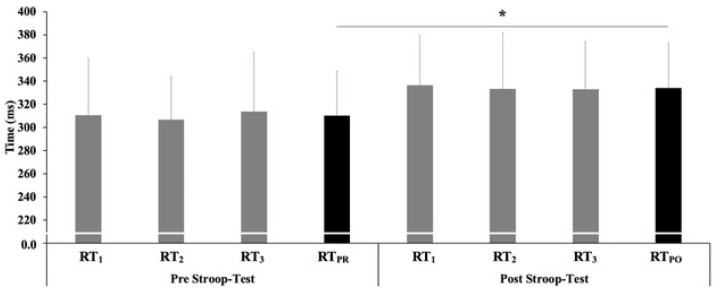
Pre-/post-Stroop test reaction time (single tests RT_1,2,3,_ in grey color). Average (in black color) RT_PR_ pre and RT_PO_ post stroop-test comparison (*—*p* < 0.05). RT_1_: Reaction time test 1 of 3; RT_2_: Reaction time test 2 of 3; RT_3_: Reaction time test 3 of 3; RT_PR_: Reaction media in the three tests in pre-test; RT_PO_: Reaction media in the three tests in post-test.

**Table 1 ijerph-19-14360-t001:** Mean and SD of hearth rate (HR) and reaction time (RT) for men and women.

	All Participants (n = 64)	Male (n = 34)	Female (n = 30)	*p* (Male × Female)
Heart Rate				
HR_MAX_ (bpm)	185.8 ± 4.3	187.6 ± 2.2	184.3 ± 5.1	0.006
T_0_: HR_R_ (%HR_MAX_)	35.1 ± 5.4	34.9 ± 6.0	35.2 ± 4.9	0.603
T_1_: HR_PRTR_ (%HR_MAX_)	38.6 ± 7.6 *	37.8 ± 8.1	39.4 ± 7.2 *	0.250
T_2_: (%HR_MAX_)	37.0 ± 5.7 *^†^	36.8 ± 6.7	37.2 ± 4.8 *^†^	0.406
T_3_: (%HR_MAX_)	40.2 ± 6.4 *	40.0 ± 6.9 *	40.4 ± 5.9 *	0.758
T_4_: (%HR_MAX_)	42.5 ± 7.1 *^†^	42.0 ± 7.5 *^†^	42.9 ± 6.8 *	0.559
T_5_: HR_POTR_ (%HR_MAX_)	38.5 ± 7.1 *	39.0 ± 7.8 *	38.1 ± 6.5 *	0.831
T_6_: (%HR_MAX_)	37.5 ± 6.7 *	37.6 ± 7.6 *	37.3 ± 5.9 ^†^	0.998
T_7_: (%HR_MAX_)	37.4 ± 6.0 *	37.6 ± 6.9 *	37.2 ± 5.3 ^†^	0.352
T_8_: (%HR_MAX_)	37.0 ± 6.6 *	37.3 ± 7.9 *	36.8 ± 5.4 ^†^	0.169
Reaction Time				
RT_PR_ (ms)	308.3 ± 40.1	316.5 ± 39.9	301.1 ± 39.4	0.410
RT_PO_ (ms)	334.1 ± 40.7 ^§^	344.3 ± 39.9 ^§^	325.1 ± 39.9 ^§^	0.067

Note: HR_MAX_: maximum heart rate; HR_R_: resting heart rate; HR_PRTR_: Heart rate after reaction time before Stroop test; HR_POTR_: Heart rate after reaction time after Stroop test; T0: at the initial trial; T1: at the end of the reaction test; T2: after 200 questions from the Stroop test; T3: after 400 questions from the Stroop test; T4: at the end of the Stroop test; T5: at the end of the reaction test; T6: at rest, after 1’; T7: at rest, after 3’; T8: at rest, after 5’; RT_PR_: reaction time before the Stroop test; RT_PO_: reaction time after Stroop test. *: represent significant difference compared with T0 (*p* < 0.05). †: represent significant difference compared with T1 (*p* < 0.05). ^§^: represent significant difference on reaction time (RT) between pre (RT_PR_) and post (RT_PR_) Stroop test (*p* < 0.05).

## Data Availability

The data that support the findings of this study are available from the corresponding author, upon reasonable request.

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
