# Peer review of "Effects of Mental Fatigue on Reaction Time in Sportsmen"

_ijerph, 2022, doi:10.3390/ijerph192114360_

Round 1

Reviewer 1 Report

Abstract.

Line 8. RT was (not were).

Line 11. How many conditions do you have? Because you indicate that there are a Stroop condition and OTHERS. You should indicate it previously.

Introdution.

The first phrase seems to be inadequate. In general, the introduction did not reflect the state of the problem. There are a lot of phrases without references (Are these your own views?). Please, explain to the readers what is the problem, focus it on a context, explain possible practical applications derived from..

Method

Familiarization day is not a experimental day. Have you got only one condition?

Results seems to be good explained, but you should think about if the design is appropriate.

References

The number of references is very low. Indeed, there are important references that does not appear in the manuscript: Van Cutsem, Bart Roelands, or Suzzy Russell are highlighted researchers that may improve the quality of your introduction and discussion. 

Author Response

Dear Editor and reviewers,

We thank you for the useful comments and suggestions to our manuscript that has now been revised accordingly. Here below you will find the answers to your comments (page and line numbers refer to the revised version of this manuscript). The changes introduced in the manuscript are highlighted with red color letter.

Reviewer 1:

Abstract

  • Line 8. RT was (not were).

Thanks. We modified it accordingly.

  • Line 11. How many conditions do you have? Because you indicate that there are a Stroop condition and OTHERS. You should indicate it previously.

Thank you. We modified the sentence.

Introduction

  • The first phrase seems to be inadequate. In general, the introduction did not reflect the state of the problem.
  • There are a lot of phrases without references (Are these your own views?).
  • Please, explain to the readers what is the problem, focus it on a context, explain possible practical applications derived from.

We are grateful for the comments. We removed the first phrase, making the rationale closer to the study’s question. And, in general, the introduction was altered through adding more references to support the information presented. In addition, we added information about the state of the problem.

Method

  • Familiarization day is not an experimental day. Have you got only one condition?

Thanks, we have changed the sentence replacing experimental day by visit. Yes, we have just one condition.

  • Results seems to be good explained, but you should think about if the design is appropriate.

We appreciate your suggestions and concerns raised at this point. We think that the explanation was unclear and, thus, we have rewritten the procedures of data collection. We modified the presentation of the results, aiming to make it more robust and clearer.

References

  • The number of references is very low. Indeed, there are important references that does not appear in the manuscript: Van Cutsem, Bart Roelands, or Suzzy Russell are highlighted researchers that may improve the quality of your introduction and discussion.

We thank you for the important comment and for the suggested references. We expanded the number of citations and added recommendations as required by you.

Reviewer 2 Report

Congratulations on your excellent work. I believe that this study has been designed and executed amazingly well. My only concern is the way you describe your statistical analysis and results. Please write again more precise and clear what you compared and how. Also, more visual will be helpful for the readers in the methodology. All the best with your manuscript.

Author Response

Dear Editor and reviewers,

We thank you for the useful comments and suggestions to our manuscript that has now been revised accordingly. Here below you will find the answers to your comments (page and line numbers refer to the revised version of this manuscript). The changes introduced in the manuscript are highlighted with red color letter.

Reviewer 2:

  • Congratulations on your excellent work. I believe that this study has been designed and executed amazingly well. My only concern is the way you describe your statistical analysis and results. Please write again more precise and clearer what you compared and how. Also, more visual will be helpful for the readers in the methodology. All the best with your manuscript.

We are grateful for the comments and pleased to meet expectations. We appreciate your recommendation. We have modified the text of the statistical analysis and the results in order to make them clearer. In addition, we have added a new figure to the methodological description to present it in a more understandable way.